# Orbital angular momentum multiplexed deterministic all-optical quantum teleportation

Shengshuai Liu [1,4], Yanbo Lou [1,4] & Jietai Jing [1,2,3✉]

Quantum teleportation is one of the most essential protocol in quantum information. In addition to increasing the scale of teleportation distance, improving its information transmission capacity is also vital importance for its practical applications. Recently, the orbital angular momentum (OAM) of light has attracted wide attention as an important degree of freedom for realizing multiplexing to increase information transmission capacity. Here we show that by utilizing the OAM multiplexed continuous variable entanglement, 9 OAM multiplexed channels of parallel all-optical quantum teleportation can be deterministically established in experiment. More importantly, our parallel all-optical quantum teleportation scheme can teleport OAM-superposition-mode coded coherent state, which demonstrates the teleportation of more than one optical mode with fidelity beating the classical limit and thus ensures the increase of information transmission capacity. Our results open the avenue for deterministically implementing parallel quantum communication protocols and provide a promising paradigm for constructing high-capacity all-optical quantum communication networks.

[1] State Key Laboratory of Precision Spectroscopy, Joint Institute of Advanced Science and Technology, School of Physics and Electronic Science, East China Normal University, 200062 Shanghai, China. [2] Department of Physics, Zhejiang University, 310027 Hangzhou, China. [3] Collaborative Innovation Center of Extreme Optics, Shanxi University, 030006 Taiyuan, Shanxi, China. [4]These authors contributed equally: Shengshuai Liu, Yanbo Lou. ✉email: jtjing@phy.ecnu.edu.cn

Quantum teleportation allows an unknown quantum state to do disembodied transportation from one location to another with the help of shared quantum entanglement and classical communications. Since the concept of quantum teleportation was proposed in 1993[1], it has attracted great attentions and achieved remarkable progresses, including both theoretical[2–4] and experimental[5–21] advances. By utilizing the quantum teleportation, various quantum information protocols have been implemented, such as entanglement swapping[22], quantum repeater[23] and quantum computing[24]. While much efforts have been devoted to increase the scale of quantum teleportation distance[14–16], its information transmission capacity, another important measure for its performance, is omitted.

Multiplexing can largely improve the information transmission capacity of optical communication system by integrating several channels into one. Multiplexing has been experimentally realized in both classical and quantum systems by using different degrees of freedom (dofs) of light, such as wavelength[25–28], polarization[29,30], space[31,32], and time[33,34]. It is well-known that the orbital angular momentum (OAM) of light[35] could support infinite number of optical modes in principle. Due to this advantage, the OAM of light[35] has attracted wide attention as an important dof for realizing multiplexing. Recently, OAM multiplexing has been experimentally realized in free space[36–39], optical fibers[40] and on chip[41].

In this article, by utilizing the OAM multiplexed continuous variables (CV) entanglement and OAM mode-matched parametric amplifier (PA) based on the four-wave mixing process (FWM)[39,42–45], we experimentally demonstrate a 9-channel parallel all-optical quantum teleportation[4]. Different from the well-known CV teleportation protocol[3,6] which requires Bell-type measurement and feed-forward techniques in classical channels, the all-optical teleportation (AOT) protocol is measurement-free and its classical channel is all-optical[4]. These features guarantee that OAM multiplexing technology in optical communication[36–41] can be directly applied in AOT. More importantly, we show that our OAM multiplexed quantum AOT scheme can teleport OAM-superposition-mode coded coherent state with fidelity beating the classical limit, which demonstrates the teleportation of more than one optical mode and thus ensures the increase of information transmission capacity[36,46].

## Results

**OAM multiplexed AOT architecture.** Figure 1 shows the scheme for parallel AOT by multiplexing OAM channels. Our OAM multiplexed Einstein-Podolsky-Rosen (EPR) entangled state[39] is generated from the double-$\Lambda$ configuration FWM

process. Since the pump beam is very strong in this FWM process, it can be regarded as classical field. Therefore, the pump noise will not add extra noise into the system. In this way, the interaction Hamiltonian for the OAM-multiplexed FWM process can be described as

$$\hat{H} = i\hbar \Sigma_\ell \gamma_\ell \hat{b}_{1,\ell}^\dagger \hat{b}_{2,-\ell}^\dagger + h.c., \tag{1}$$

where $\hat{b}_{1,\ell}^\dagger$ and $\hat{b}_{2,-\ell}^\dagger$ are the creation operators associated with an OAM mode in EPR$_1$ and EPR$_2$ beams, respectively. $\ell$ and $-\ell$ are the corresponding topological charges. $h.c.$ is the Hermitian conjugate. $\gamma_\ell$ is the interaction strength of each pair of OAM modes in FWM process. Based on Eq. (1), each input-output relation of the OAM-multiplexed FWM process can be expressed as

$$\hat{b}_{1,\ell} = \sqrt{G_{1,\ell}}\hat{b}_{in1,\ell} + \sqrt{G_{1,\ell} - 1}\hat{b}_{in2,-\ell}^\dagger, \tag{2}$$

$$\hat{b}_{2,-\ell}^\dagger = \sqrt{G_{1,\ell} - 1}\hat{b}_{in1,\ell} + \sqrt{G_{1,\ell}}\hat{b}_{in2,-\ell}^\dagger, \tag{3}$$

where $G_{1,\ell} = \cosh^2(\gamma_\ell \tau) = \cosh^2(\gamma_{-\ell} \tau)$ is the FWM gain of the corresponding OAM mode. $\tau$ is the time scale of interaction. $\hat{b}_{in1,\ell}$ and $\hat{b}_{in2,-\ell}$ are vacuum inputs. Then, for each OAM-multiplexed channel, $\hat{b}_{2,-\ell}$ is distributed to Alice (sender), and $\hat{b}_{1,\ell}$ is distributed to Bob (receiver). Alice amplifies the input state $\hat{a}_{in,\ell}$ carrying OAM mode through an OAM mode-matched PA[39] with the help of $\hat{b}_{2,-\ell}$. The intensity gain of the corresponding OAM-mode coded $\hat{a}_{in,\ell}$ is $G_{2,\ell}$. When the pump beam is a Gaussian beam, this PA can match its input states with opposite topological charges of $\ell$ and $-\ell$ due to OAM conservation. After such amplification, $\hat{a}_{c,\ell}$ (i.e., the amplified $\hat{a}_{in,\ell}$) can be expressed as

$$\hat{a}_{c,\ell} = \sqrt{G_{2,\ell}}\hat{a}_{in,\ell} + \sqrt{G_{2,\ell} - 1}\hat{b}_{2,-\ell}^\dagger. \tag{4}$$

According to the definition given in other works[4,47], this OAM mode-matched PA based on double-$\Lambda$ configuration FWM process[45], as described by Eq. (4), is a linear amplifier. Then, Alice sends $\hat{a}_{c,\ell}$ to Bob through an all-optical classical channel. To retrieve the input state, receiver Bob couples $\hat{a}_{c,\ell}$ and $\hat{b}_{1,\ell}$ by a beam splitter with a transmission of $\varepsilon = 1/G_{2,\ell}$. Then, the output

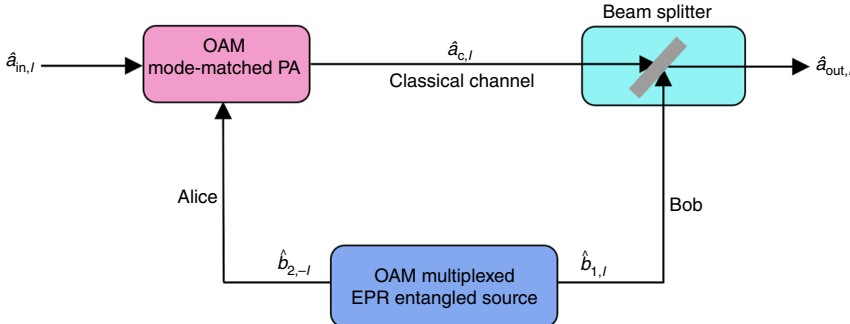

**Fig. 1 Schematic of parallel all-optical teleportation.** Alice, sender; Bob receiver; OAM orbital angular momentum, EPR entangled source Einstein-Podolsky-Rosen entangled source; PA parametric amplifier, $\hat{a}_{in,\ell}$, input state for one OAM-multiplexed channel; $\hat{a}_{c,\ell}$, amplified $\hat{a}_{in,\ell}$; $\hat{a}_{out,\ell}$, retrieved state for one OAM-multiplexed channel; $\hat{b}_{1,\ell}$ and $\hat{b}_{2,-\ell}$ are the annihilation operators associated with an OAM mode in EPR$_1$ and EPR$_2$ beams, respectively.

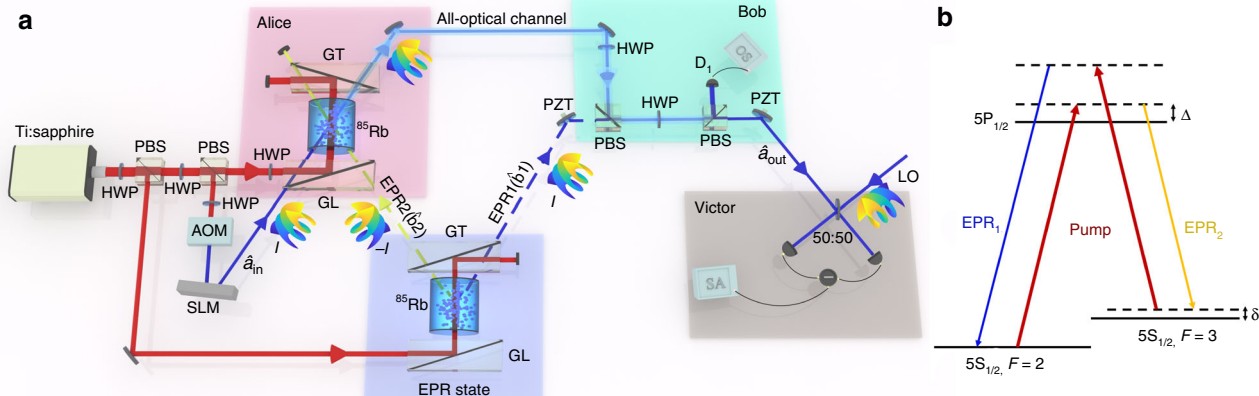

**Fig. 2 Detailed experimental setup for parallel all-optical teleportation. a** Alice, Bob, and Victor act as sender, receiver, and verifier of the all-optical teleportation (AOT), respectively. Alice and Bob are connected by an all-optical channel. $\hat{a}_{in}$ input state populated by a coherent state $|\alpha_{in}\rangle$, EPR state Einstein-Podolsky-Rosen entangled state, $^{85}$Rb vapor cell, HWP half wave plate, PZT piezoelectric transducer, PBS polarization beam splitter, AOM acousto-optic modulator, SLM spatial light modulator, GL Glan-Laser polarizer, GT Glan-Thompson polarizer, $\hat{a}_{out}$ retrieved state, $D_1$ photodetector, OS oscilloscope, 50:50 50:50 beam splitter, SA spectrum analyzer. The resolution bandwidth (RBW) of the SA is 1 MHz. The video bandwidth (VBW) of the SA is 100 Hz. **b** Energy level diagram of $^{85}$Rb D1 line for four-wave mixing (FWM) process in our scheme. $\Delta$ one-photon detuning, $\delta$ two-photon detuning.

state ($\hat{a}_{out,\ell}$) retrieved by Bob reads

$$
\begin{aligned}
\hat{a}_{out,\ell} &= \hat{a}_{in,\ell} + \sqrt{\frac{G_{2,\ell}-1}{G_{2,\ell}}}(\hat{b}_{2,-\ell}^{\dagger} - \hat{b}_{1,\ell}) \\
&\approx \hat{a}_{in,\ell} + \hat{b}_{2,-\ell}^{\dagger} - \hat{b}_{1,\ell},
\end{aligned}
\tag{5}
$$

under the approximation $G_{2,\ell} \gg 1$. The two last terms in Eq. (5) vanish in the limit of an EPR entangled state ($G_{1,\ell} \gg 1$) and very high parametric gain ($G_{2,\ell} \gg 1$), leaving as an output state only the first term, which is equal to the input state $\hat{a}_{in,\ell}$, thus we achieve teleportation. In other words, although the amplification process will certainly add noise to the input state as indicated by Eq. (4), such added noise will be greatly reduced by the appropriate attenuation ratio of the beam splitter at Bob station and the coupling of two entangled EPR beams on the same beam splitter as shown by Eq. (5). In this way, the noise cancellation is realized by using the quantum correlation between the two entangled EPR beams, thus ensuring the realization of quantum teleportation. In a word, the noise introduced by this amplification process is indispensable for realizing quantum AOT. It plays the role of introducing $EPR_2$ into the system for implementing the subsequent noise cancellation operation at Bob station, then makes the fidelity beating the classical limit and thus realizing the quantum AOT.

In teleportation, the quality of the retrieved state can be quantified by its teleportation fidelity $F \in [0, 1]$. If the fidelity $F$ is higher than the classical limit of AOT without the help of entanglement, the quantum teleportation succeeds. The fidelity $F$ is defined as the overlap between the input state and the output state characterized by the density matrix. For a coherent state carrying OAM, the fidelity can be expressed as[48]

$$
F_\ell = \frac{2}{\sigma_Q} \exp\left[ -\frac{2}{\sigma_Q} |\beta_{out} - \beta_{in}|^2 \right],
\tag{6}
$$

where $\sigma_Q$ is the variance of the teleported state in representation of the Q function and $\sigma_Q = \sqrt{\left(1 + \sigma_W^{x_\ell}\right)\left(1 + \sigma_W^{p_\ell}\right)}$. $\sigma_W^{x_\ell}$ ($\sigma_W^{p_\ell}$) is the fluctuation variance of amplitude (phase) quadrature. $\beta_{in}$ and $\beta_{out}$ are the amplitudes of the input state at Alice station and the output state at Bob station, respectively. Based on Eq. (5), $\sigma_W^{x_\ell}$ and

$\sigma_W^{p_\ell}$ can be given by

$$
\sigma_W^{x_\ell} = \sigma_W^{p_\ell} = 1 + \frac{2(G_{2,\ell}-1)}{G_{2,\ell}}\left(\sqrt{G_{1,\ell}} - \sqrt{G_{1,\ell}-1}\right)^2.
\tag{7}
$$

Then, the fidelity of deterministic quantum AOT for a coherent input state carrying OAM can be expressed as

$$
\begin{aligned}
F_\ell &= \frac{1}{1 + \frac{G_{2,\ell}-1}{G_{2,\ell}}\left(\sqrt{G_{1,\ell}} - \sqrt{G_{1,\ell}-1}\right)^2} \\
&\approx \frac{1}{1 + \left(\sqrt{G_{1,\ell}} - \sqrt{G_{1,\ell}-1}\right)^2},
\end{aligned}
\tag{8}
$$

under the approximation $G_{2,\ell} \gg 1$.

Our detailed experimental setup for parallel AOT by multiplexing OAM channels is shown in Fig. 2a. A cavity stabilized Ti:sapphire laser emits a Gaussian laser beam. Its frequency is 0.9 GHz blue detuned from the $^{85}$Rb D1 line ($5S_{1/2}$, $F = 2 \rightarrow 5P_{1/2}$). A polarization beam splitter (PBS) is used to divide the laser into two. One beam, which is vertically polarized and has a power of 100 mW, is served as the pump beam of the FWM process in a $^{85}$Rb vapor cell for producing OAM multiplexed EPR entangled state[39]. Such pump power is strong enough to maximize the squeezing of our system. As shown in Fig. 2b, in this double-$\Lambda$ configuration FWM process, two pump photons convert to one photon for $EPR_1$ (blue-shifted from the pump beam) and one photon for $EPR_2$ (red-shifted from the pump beam). Reflected by a Glan-Laser polarizer (GL), this pump beam is seeded into the $^{85}$Rb vapor cell which is 12 mm long and stabilized at 113 °C. This pump beam has a large waist of about 930 μm at the center of vapor cell. This pump beam with strong enough power of 100 mW and large waist of 930 μm ensures a sufficient number of OAM multiplexed quantum AOT channels[39]. The residual pump beam after the FWM process is eliminated by a Glan-Thompson polarizer (GT) with an extinction ratio of $10^5$:1. In this way, we obtain the OAM-multiplexed EPR entangled state for realizing quantum AOT. After that, the $EPR_2$ is distributed to Alice (sender), and $EPR_1$ is distributed to Bob (receiver). The other beam from the first PBS, which is horizontally polarized, is also divided into two by another PBS. The weak one passes through an acousto-optic modulator

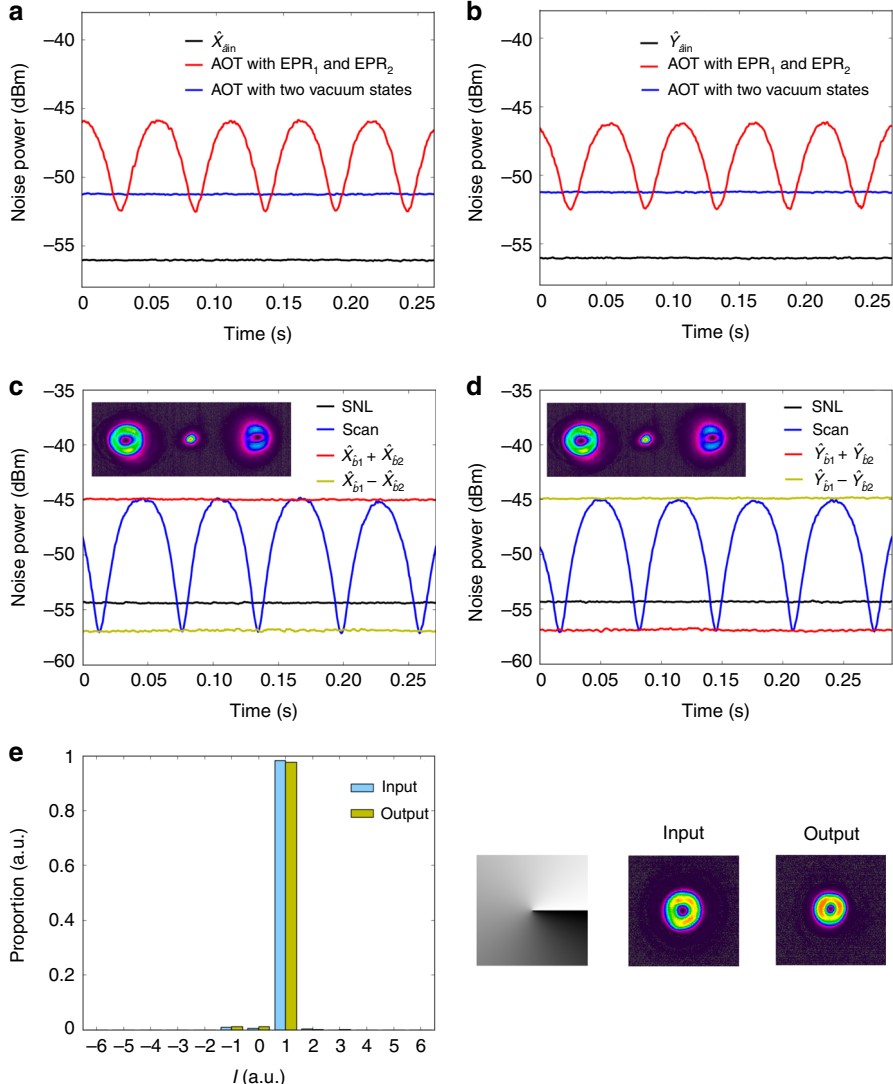

**Fig. 3 The quadrature variances of the retrieved state for $\ell = 1$. a, b** The variance of amplitude (phase) quadrature of the retrieved state. **c, d** The amplitude (phase) quadrature measurement of the orbital angular momentum (OAM) multiplexed entanglement source. The yellow trace is the variance of amplitude (phase) quadrature difference. The red trace is the variance of amplitude (phase) quadrature sum. The blue curve is noise power of the photocurrents output from balanced homodyne detection (BHD) versus the scanning phase. The black trace represents the corresponding shot noise limit (SNL). The insets show the transverse pattern of the output field when the four-wave mixing (FWM) is seeded by a bright beam. The left, center, and right beams are $\hat{b}_{1,1}$, Gaussian pump beam, and $\hat{b}_{2,-1}$, respectively. **e** The results of OAM mode analysis. The computer-generated hologram for $\ell = 1$ case and the corresponding images of input and output fields are shown on the right inset, respectively.

(AOM) and a spatial light modulator (SLM) to generate the OAM-mode coded input state $\hat{a}_{\text{in}}$ which is blue-shifted about 3.04 GHz from the pump beam and has a power of 0.4 μW. The strong one is served as the pump beam of OAM mode-matched PA at Alice station. This OAM mode-matched PA is also based on the double-$\Lambda$ configuration FWM process in another $^{85}$Rb vapor cell which is 12 mm long and stabilized at 110 °C. Alice amplifies the input state $\hat{a}_{\text{in}}$ carrying OAM modes through this OAM mode-matched PA with the help of EPR$_2$. Combined by a GL, the pump, input state $\hat{a}_{\text{in}}$, and EPR$_2$ are crossed in the center of the $^{85}$Rb vapor cell. Due to OAM conservation, only the input state $\hat{a}_{\text{in}}$ and EPR$_2$ with opposite topological charges can be coupled when the pump is a Gaussian beam. The angle between input state $\hat{a}_{\text{in}}$ and EPR$_2$ is about 14 mrad and the pump beam is symmetrically crossed with $\hat{a}_{\text{in}}$ and EPR$_2$ beams in the same plane. We set the intensity gain of $\hat{a}_{\text{in}}$ at $10 \gg 1$, which ensures that the amplified $\hat{a}_{\text{in}}$ can be regarded as a

classical field[4]. The residual pump beam after this mode-matched PA is eliminated by another GT. After such amplification, Alice sends the amplified $\hat{a}_{\text{in}}$ to Bob through an all-optical classical channel. To retrieve the input state, receiver Bob couples the amplified $\hat{a}_{\text{in}}$ and EPR$_1$ by a PBS and then attenuates this coupled beam with a beam splitter which consists of a half wave plate (HWP) and another PBS. Here, a piezo-electric transducer (PZT) is placed in the path of EPR$_1$ to change the relative phase between EPR$_1$ and EPR$_2$. The transmission of this beam splitter is $\varepsilon = 1/10$, which can be precisely set by the photodetector (D$_1$) connecting to an oscilloscope (OS). Then, Victor measures the amplitude (phase) quadrature variance of the teleported OAM mode ($\hat{a}_{\text{out}}$) by balanced homodyne detection (BHD) and calculates its teleportation fidelity $F$. Another PZT is placed in the path of $\hat{a}_{\text{out}}$ to change the relative phase between $\hat{a}_{\text{out}}$ and local oscillator (LO). The LO beam with a power of 650 μW is

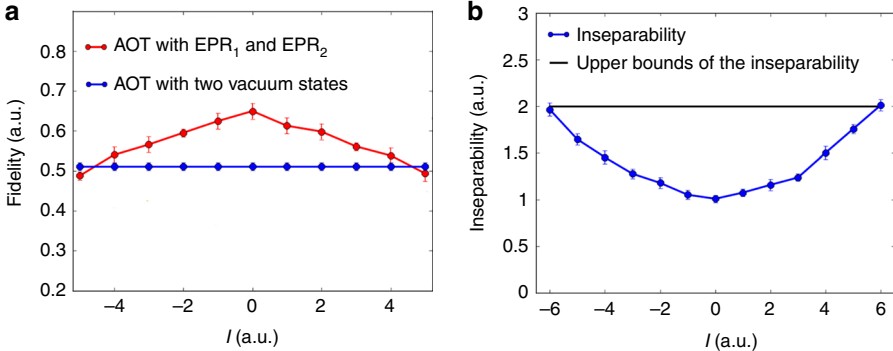

**Fig. 4 Fidelity of all-optical teleportation versus the topological charge of the input modes. a** The fidelity of all-optical teleportation (AOT) for teleporting different orbital angular momentum (OAM) modes. The red dot trace indicates the fidelity of AOT with the help of EPR$_1$ and EPR$_2$. The blue dot trace indicates the fidelity of AOT with only two vacuum states. **b** The inseparability of the OAM multiplexed entanglement source versus topological charge $\ell$ (the blue dot trace). The black straight line denotes the upper bound of the inseparability. The error bars are obtained from the standard deviations of multiple repeated measurements.

obtained by setting up a similar FWM process in the OAM-mode matched PA at Alice station, which is a few mm above the current beams[43]. Its pump is split from the pump beam of OAM-mode matched PA, while its seed is split from the beam right after the SLM and before $\hat{a}_{in}$. In this way, the frequency of LO beam naturally matches the one of $\hat{a}_{out}$. Our balanced homodyne detector has a transimpedance gain of $10^5$ V A$^{-1}$ and a quantum efficiency of 97%.

**Teleporting a single OAM mode**. The amplitude and phase quadrature variances of the teleported state carrying OAM mode with $\ell = 1$ measured by Victor's BHD at 2 MHz sideband are shown in Fig. 3a (locking the relative phase between LO beam and $\hat{a}_{out}$ to 0 by micro-control unit[49,50]) and Fig. 3b (locking the relative phase between LO beam and $\hat{a}_{out}$ to $\pi/2$ by proportional-integral-differential circuit[49]), respectively. The variance of amplitude quadrature difference $\hat{X}_{\hat{b}_{1,1}} - \hat{X}_{\hat{b}_{2,-1}}$ (phase quadrature sum $\hat{Y}_{\hat{b}_{1,1}} + \hat{Y}_{\hat{b}_{2,-1}}$) of the EPR entanglement is $2.51 \pm 0.16$ dB ($2.58 \pm 0.18$ dB) below the corresponding shot noise limit (SNL) as shown in Fig. 3c (Fig. 3d), indicating the presence of EPR entanglement between these two OAM modes. As shown in Fig. 3a, b, the black traces, measured by blocking the EPR entanglement and pump beams, are the quadrature variances of the input state. To realize AOT, the EPR entanglement is distributed to Alice and Bob. The relative phase between EPR$_1$ and EPR$_2$ is scanned by a PZT. The measured quadrature variances of the retrieved state by Victor are shown as the red traces in Fig. 3a [amplitude quadrature ($\hat{X}_{\hat{a}_{out}}$)] and Fig. 3b [phase quadrature ($\hat{Y}_{\hat{a}_{out}}$)]. The minima of red trace for the variance of amplitude quadrature (phase quadrature) indicates that the relative phase between EPR$_1$ and EPR$_2$ corresponds to $\hat{X}_{\hat{b}_{1,1}} - \hat{X}_{\hat{b}_{2,-1}}$ ($\hat{Y}_{\hat{b}_{1,1}} + \hat{Y}_{\hat{b}_{2,-1}}$). Therefore, we can treat the minima of each red trace as the variance of $\hat{X}_{\hat{a}_{out}}$ or $\hat{Y}_{\hat{a}_{out}}$ quadrature of the output retrieved state under the help of EPR entanglement. Both the variances of $\hat{X}_{\hat{a}_{out}}$ and $\hat{Y}_{\hat{a}_{out}}$ quadratures are almost equal and $3.58 \pm 0.21$ dB above the corresponding quadrature variances of the input state. This indicates that the corresponding fidelity of the retrieved state by quantum AOT with the help of EPR entanglement is $0.61 \pm 0.02$. To have the corresponding classical AOT for comparison, we block the EPR entanglement. The measured quadrature variances are shown as blue traces in Fig. 3a and b, which are $4.66 \pm 0.12$ dB above the corresponding quadrature

variances of the input state. This gives a fidelity of $0.51 \pm 0.01$ for such classical AOT as the classical limit. Hence, it shows that such classical limit is beaten by the quantum AOT with the help of EPR entanglement. In addition, we find that the OAM mode carried by the retrieved state is almost the same with input state as shown in Fig. 3e.

**OAM multiplexed teleportation channels**. To illustrate the parallelism of our OAM multiplexed AOT, we do a series of the above measurements while changing the topological charge $\ell$ carried by an input state from $-5$ to 5, and keeping $G_{2,\ell} = 10$. As shown in Fig. 4a, the red dot trace is the fidelity of quantum AOT with EPR entanglement for teleporting different OAM modes, while the blue dot trace is the fidelity of classical AOT with only two vacuum states. It shows that the fidelity of quantum AOT for topological charge in the range of $-4$ to 4 is better than the corresponding fidelity of the classical AOT. In addition, we can see that for the quantum AOT with EPR entanglement, the larger the $|\ell|$ is, the lower the fidelity will be. This is because the inseparability of OAM-multiplexed entanglement source gets worse with the increase of $|\ell|$ as shown in Fig. 4b. We calculate the inseparability of OAM-multiplexed entanglement source by $I_{\hat{b}_{1,\ell},\hat{b}_{2,-\ell}} = Var(\hat{X}_{\hat{b}_{1,\ell}} - \hat{X}_{\hat{b}_{2,-\ell}}) + Var(\hat{Y}_{\hat{b}_{1,\ell}} + \hat{Y}_{\hat{b}_{2,-\ell}})$[51–53], which is directly related to the two-mode squeezing of OAM multiplexed entanglement. The corresponding mode analyses for the input and retrieved states are shown in Fig. 5. These results clearly show that we have successfully constructed 9 OAM multiplexed channels of parallel quantum AOT with fidelities beating the classical limit.

To directly show the advantage of our OAM multiplexed AOT scheme in terms of increasing the information transmission capacity, we implement the teleportation of OAM-superposition-mode coded coherent state ($\hat{a}_{in} = \hat{a}_{in,\ell} + \hat{a}_{in,-\ell}, \ell = 1, 2, 3, 4$). Firstly, in order to prove that the OAM superposition mode carried by input coherent state remains unchanged after AOT, we make a series of OAM mode analyses of the input and the retrieved output states as shown in Fig. 6. It shows that our AOT can well maintain the OAM superposition mode distribution of the input states. Secondly, we measure the fidelity of quantum AOT for teleporting OAM-superposition-mode coded coherent state. For $\hat{a}_{in} = \hat{a}_{in,\ell} + \hat{a}_{in,-\ell}, \ell = 1, 2, 3, 4$, the fidelity of such quantum AOT as indicated on the top of each subfigure is better than the measured fidelity of corresponding classical AOT ($0.51 \pm 0.01$). In principle, the fidelity for

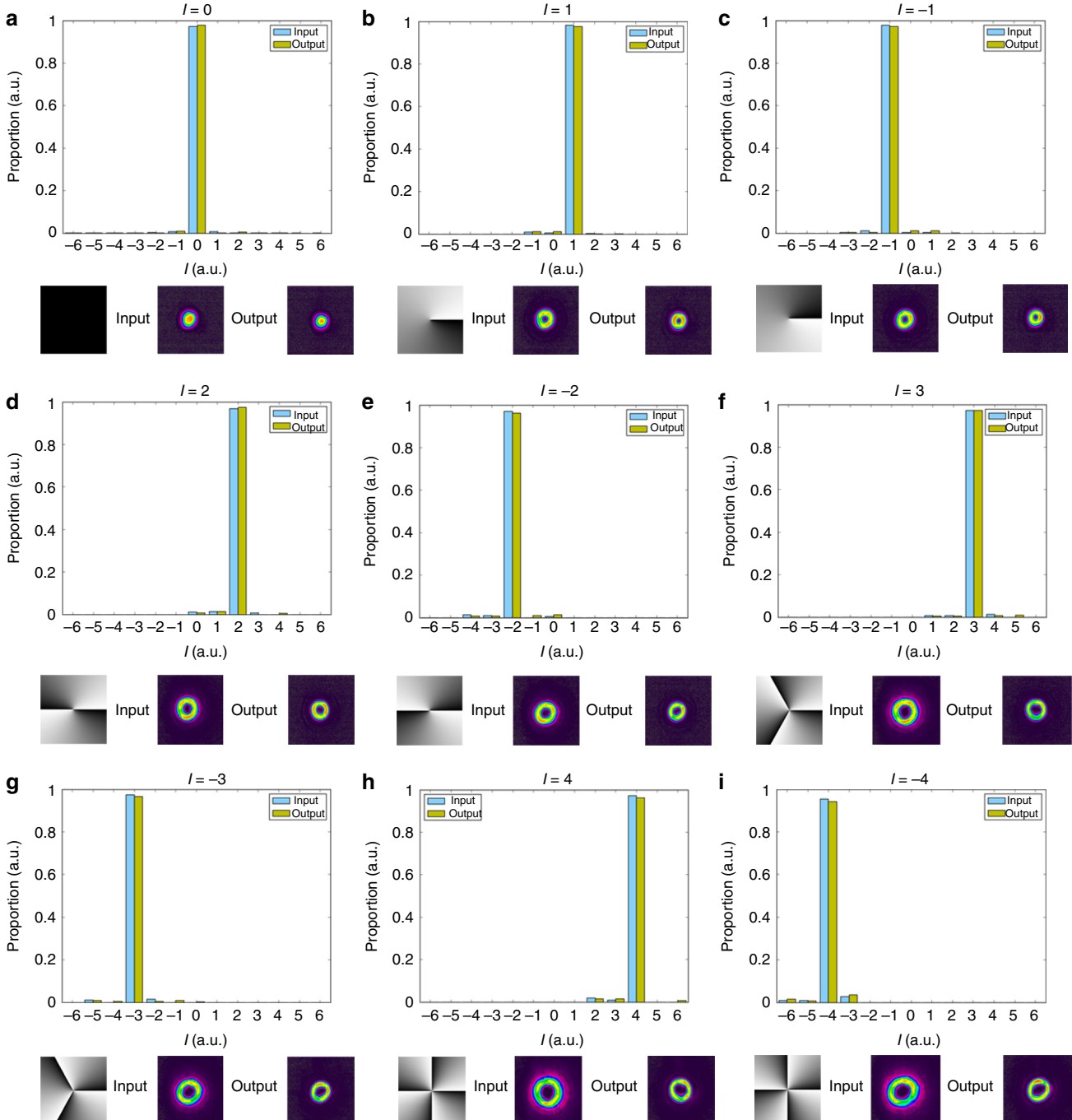

**Fig. 5 Orbital angular momentum mode analysis for the input and output fields.** The bottom row in each subfigure shows the computer-generated hologram for OAM mode and the corresponding images of input and output fields.

teleporting $\hat{a}_{in} = \hat{a}_{in,\ell} + \hat{a}_{in,-\ell}$ is same as the one for teleporting $\hat{a}_{in} = \hat{a}_{in,\ell}$ ($\hat{a}_{in,-\ell}$) because the squeezing levels of EPR entanglement for these two cases are equal[39]. In our experiment, the slight difference between the fidelities for these two cases is caused by measurement error. These results clearly illustrate that our OAM multiplexed AOT can simultaneously teleport more than one OAM mode with fidelity beating the classical limit.

## Discussion
Although the OAM modes themselves are a subset of a discrete Hilbert space, what we teleported is the amplitude quadrature and phase quadrature of the coherent state, which are described by a continuous Hilbert space. From this point of view, our experiment demonstrates 9 parallel channels of deterministic CV quantum teleportation of coherent states carrying different OAM modes. Moreover, we have shown that our OAM multiplexed AOT scheme can teleport OAM-superposition-mode coded coherent state, which implements the simultaneous teleportation of more than one optical mode. The successful implementation of such OAM multiplexed quantum teleportation benefits from the measurement-free AOT architecture, which makes it possible to implant the OAM multiplexing technique of the modern optical communication into its all-optical channels. Our results open the avenue for deterministically implementing parallel quantum communication protocols and provide a promising paradigm for

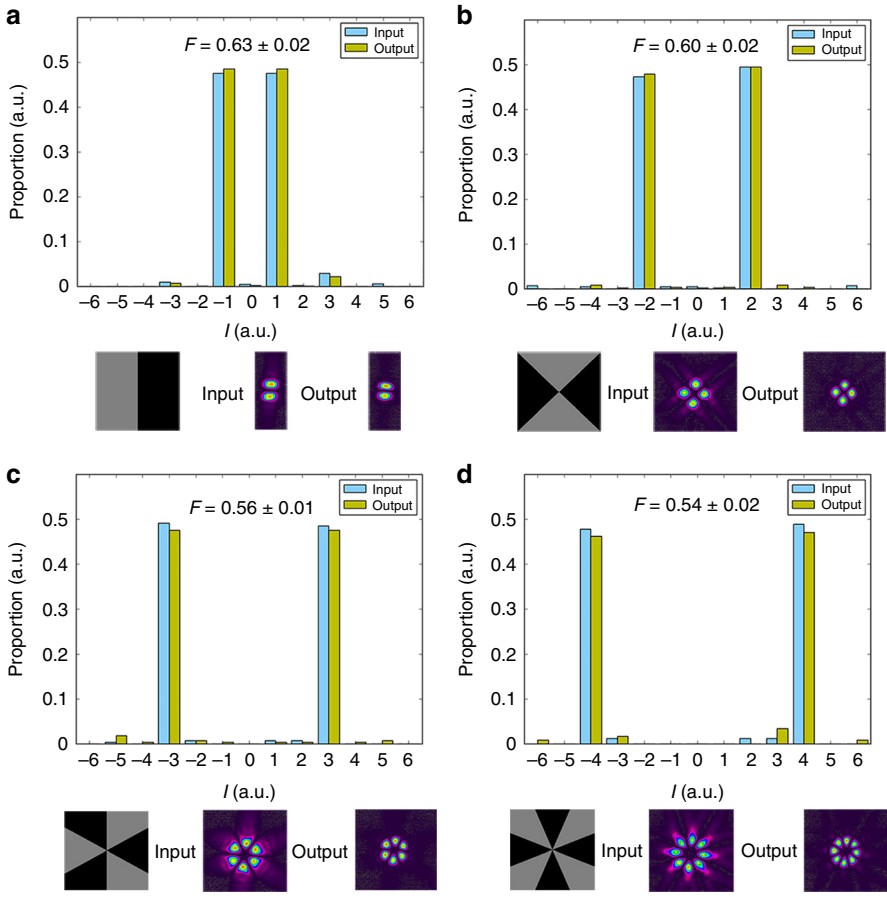

**Fig. 6 Teleporting orbital angular momentum superposition mode $\hat{a}_{\mathbf{in}} = \hat{a}_{\mathbf{in},\ell} + \hat{a}_{\mathbf{in},-\ell}$.** The distribution of orbital angular momentum (OAM) superposition mode before and after all-optical teleportation (AOT) when **a** $\ell = 1$, **b** $\ell = 2$, **c** $\ell = 3$ and **d** $\ell = 4$. The corresponding AOT fidelities are 0.63, 0.60, 0.56, 0.54, respectively. The bottom row in each subfigure shows the computer-generated hologram for OAM superposition mode and the corresponding images of input and output fields. The error bars are obtained from the standard deviations of multiple repeated measurements.

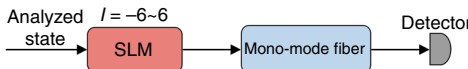

**Fig. 7 The scheme of orbital angular momentum mode analysis.** SLM spatial light modulator.

constructing high-capacity all-optical quantum communication networks.

## Methods

**The OAM mode analysis of teleported state**. We use the same method as mentioned in other work[54] to analyze the OAM mode of the state. As shown in Fig. 7, the analyzed state is sent to the SLM. Only the OAM mode carrying topological charge of $\ell$ can be transformed into a Gaussian mode by loading a computer-generated hologram with a topological charge $-\ell$ on the SLM. Due to the fact that only the Gaussian mode can be coupled into the mono-mode fiber, the output beam power of the mono-mode fiber almost only contains the power of the corresponding teleported OAM mode carrying topological charge of $\ell$ under high coupling efficiency of the fiber. We load the computer-generated hologram with a topological charge ranging from $-6$ to 6 on the SLM. In this way, we can obtain the proportion of the OAM mode with a topological charge from $-6$ to 6.

## Data availability

The data that support the findings of this study are available from the corresponding author upon reasonable request.

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

## Acknowledgements

This work was supported by the National Natural Science Foundation of China (Grants No. 11874155, No. 91436211, No. 11374104, and No. 10974057), the Natural Science Foundation of Shanghai (Grant No. 17ZR1442900), the Minhang Leading Talents (Grant No. 201971), ECNU Academic Innovation Promotion Program for Excellent Doctoral Students (Grant No. YBNLTS2020-046), the Program of Scientific and Technological Innovation of Shanghai (Grant No. 17JC1400401), the National Basic Research Program of China (Grant No. 2016YFA0302103), Shanghai Municipal Science and Technology Major Project (Grant No. 2019SHZDZX01), the 111 Project (Grant No. B12024), and the Fundamental Research Funds for the Central Universities.

## Author contributions

J.J., S.L., and Y.L. conceived the idea and designed the experiment. S.L., Y.L., and J.J. performed the experimental measurements and analytical calculations. All authors co-wrote the manuscript. J.J. supervised the whole project as group leader.

## Competing interests

The authors declare no competing interests.
