## [Peer Review File · Nature Communications]

Reviewers' Comments:

Reviewer #1:

Remarks to the Author:

The authors set out to demonstrate teleportation of states within a 9 dimensional subset of orbital angular momentum modes, following from their recent work on entanglement via FWM in an atomic vapour. This is a timely and technologically exciting idea, expanding on the first demonstration of 3D teleportation (realised as path encoding) which was published only last year by Yi-Han Luo et al. (Quantum Teleportation in High Dimensions, DOI: 10.1103/PhysRevLett.123.070505) - a paper that the authors curiously do not cite.

The "all-optical teleportation" scheme used in this manuscript is based on a theoretical proposal from 1999 which was until now not known to me. It is notably different from conventional teleportation. Rather than sending information on the measurement outcome on Alice's state together with one of the EPR twin photons, here the amplified state itself is sent via a classical channel, seeming closer to "portation" rather than "teleportation". The fidelity that can be reached in this scheme is inherently limited by vacuum noise in addition to detector noise.

While I find the experiment presented in the manuscript appealing, and potentially suitable for publication in Nature Communications, the manuscript is missing

- a clear description of the teleportation scheme,
- details on the atomic processes for the EPR pair generation and in particular the operation of the parametric amplifier,
- a more thorough theoretical analysis,
- an explanation why a continuous variable scheme is required for states described in a subset of a discrete Hilbert space,
- experimental details on the generation of the state to be teleported.

Reviewer #2:

Remarks to the Author:

The manuscript by S. Liu reports the experimental demonstration of the continuous variable teleportation protocol realised using entangled orbital angular momentum (OAM) modes of light beams. The teleportation scheme implemented is the all-optical quantum teleportation proposed by T. Ralph in 1999. Instead of Bell state measurements and feed-forward, all-optical quantum teleportation requires linear amplification of the input state. In the present experiment, the linear amplifier is replaced by an OAM mode-matched parametric amplifier. The paper shows that teleportation can be established using OAM modes up to the topological charge of 4, beating the classical fidelity limit. The teleportation of OAM mode superpositions has also been demonstrated, showing the possibility to achieve simultaneous teleportation of OAM modes.

The paper is very interesting and I support its publication in Nature Communications; however, I have some comments and questions that should be addressed before final acceptance.

- 1) In the description of the experiment (Results section), the EPR source should be pumped with a Gaussian beam but is not specified. About this I have a curiosity that could be interesting for clarifying the generation mechanism. Which is the limiting factor in using a smaller waist? Is it useful to improve the parametric gain and the squeezing?
- 2) Somehow related to the question above, the EPR source seems to be the main limitation in the maximum topological charge that can be reached for quantum teleportation as shown in Fig 3. The author should explain how the pump beam can influence the EPR correlations between OAM modes.
- 3) The local oscillator beam is not described in Fig. 1, some descriptions should also be given along with the text. How it is obtained? How much is the local oscillator power? In which way the its

phase is locked?

4) The quantum efficiency of the homodyne detector is not reported.

5) The implementation of all-optical quantum teleportation requires a linear amplifier. Instead, here a seeded noncollinear parametric amplifier based on FWM is used. Some clarification should be given along the main text. It would be useful to insert a comment on the equivalence of both approaches. In which way the fidelity is influenced by the noise added during the amplification process? Is the amplifier pump noise also a limiting factor?

6) The caption of Fig. 1: the notation $|\alpha_{in}\rangle$ (\hat{a}_{in}) is not clear, it should be replaced by mode \hat{a}_{in} populated by a coherent state $|\alpha_{in}\rangle$. Fig. 1 should be changed accordingly. There is a double Rb in the caption.

7) It is not clear how the inseparability is calculated. Some clarifications should be given together with references.

8) In Fig.2 the insets of Fig.2c and Fig.2d are not clearly described. The meaning of left, center, and right beams should be explained.

Reviewer #3:

Remarks to the Author:

The paper presents the experimental results of an all-optical (OA) teleportation protocol, where a coherent state encoded on an orbital angular momentum (OAM) mode of light ($\ell=1$) is teleported with Fidelity ~ 0.61 (beating the classical limit, i.e., $F=0.5$). Even though the achieved Fidelity is less than the corresponding one achieved through the (conventional) Braunstein-Kimble (BK) protocol, i.e., $F\sim 0.83$ [M. Yukawa et al., Phys. Rev. A 77, 022314 (2008)], this is an important result, since (to my knowledge) that's the first time that the OA teleportation protocol has been experimentally demonstrated, so further progress and higher Fidelities should be expected in the future. The results of this work can potentially make this alternative way to perform teleportation widely known to the community, and other research groups might follow up in the future in order to achieve even higher values of fidelity. Another important result of this work is the experimental demonstration of parallel AO teleportation of two coherent modes (corresponding to different topological charges of OAM), which can be used to increase the quantum capacity of quantum channels.

The manuscript is well-written and it is worth to be published in Nature Communications for the reasons mentioned before, however, there is a specific claim in the paper regarding the security of the teleportation that I would strongly advise to be removed.

In particular, the authors argue that if we block the arm of the entangled state that goes to Bob, then Bob acts as an eavesdropper (Eve), and under this modification, the new measured Fidelity ~ 0.37 lies below the classical limit ($F=0.5$), and thus they show security against eavesdropping. However, that's not the right way of assessing security in a teleportation protocol. Following the discussion of Ref. [F. Grosshans and P. Grangier, Phys. Rev. A 64, 010301(R) (2001)] (or the more recent analysis that connects the security of the teleportation with the entanglement/steering shared between Alice and Bob in Ref. [Q. He et al., Phys. Rev. Lett. 115, 180502 (2015)]), the security of the teleported state is related to Alice's ability to cheat by cloning the initial state. In particular, let us assume that Alice can perfectly perform a quantum cloning protocol, and thus achieve two clones of the initial state with Fidelities of $2/3$ each. Then in principle she can perfectly teleport one of her clones to Bob. However, since Bob doesn't know that Alice cheated, he measures $F=2/3$. Thus, in order Bob to be sure that his state is better than any other potential clones he needs to achieve a fidelity higher than $2/3$ (also known as the no-cloning

limit). Another way of looking at the same problem is by assuming that Eve has full access to the losses along at least one arm of the entangled state (along with her access to the classical channel between Alice and Bob). Then Eve's best strategy is to build her own version of the teleported state. If the transmissivity of the channel is below $1/2$ then Eve can create a better copy than Bob, which again means that $F > 2/3$ is required for a secure teleportation.

I also have a question regarding the results of the parallel OA teleportation. More specifically, in Fig.4(a) a Fidelity ~ 0.63 is reported for the superposition of modes $\ell = -1$ and $\ell = 1$. I find it really weird that this fidelity is higher than the fidelity reported for the single teleported mode $\ell = 1$, i.e., $F \sim 0.61$. I would assume that teleporting a superposition of modes is a harder task (as it is actually shown later in Fig.(b)-(d) with Fidelities ranging from ~ 0.54 to ~ 0.6). It would be nice if the authors could provide an explanation for this counter-intuitive result.

Below, I also have a list of further (minor) suggestions:

1. The first sentence in the abstract "Quantum teleportation is the most essential and fascinating protocol in quantum information." should be toned down a bit, and rewritten as "one of the most essential..."
2. Right after Eq.(2) the authors should clearly state that the two last terms in Eq.(2) vanish in the limit of an EPR entangled state and very high parametric gain, leaving as an output state only the first term, which is equal to the input state, thus we achieve teleportation.
3. In Fig.3 a comparison between the established entanglement and the achieved fidelity for the different OAM modes is presented. It is true that in the literature the separability criterion (or the logarithmic negativity, which is a monotonic function of the maximum possible violation of the separability criterion) has been extensively used to "measure" entanglement, but there are more rigorous ways to quantify entanglement, such as: (i) entanglement of formation, (ii) relative entropy of entanglement, and (iii) squashed entanglement.
4. I think that the authors should explain the reasoning behind picking a half wave plate to represent the all-optical channel between Alice and Bob.
5. The papers [M. Yukawa et al., Phys. Rev. A 77, 022314 (2008)] and [N. Lee et al., Science 332, 330 (2011)] should be cited in the introduction regarding previous works on experimental teleportation results, since they present the current state of the art on BK teleportation experiments.

Response to the Reviewers:

Replies to Reviewer 1

Report of Reviewer 1 -- NCOMMS-20-10407-T

Reviewer's Comment:

The authors set out to demonstrate teleportation of states within a 9 dimensional subset of orbital angular momentum modes, following from their recent work on entanglement via FWM in an atomic vapour. This is a timely and technologically exciting idea, expanding on the first demonstration of 3D teleportation (realised as path encoding) which was published only last year by Yi-Han Luo et al. (Quantum Teleportation in High Dimensions, DOI: 10.1103/PhysRevLett.123.070505) - a paper that the authors curiously do not cite.

Our Reply:

We thank the reviewer for carefully reading our paper and giving a concise and accurate summary of our work. We also thank the reviewer for pointing out the recent result of 3D teleportation. Our work belongs to continuous variable regime and exploits the OAM modes of optical fields for realizing multiplexing, while the work PhysRevLett.123.070505 mentioned by the reviewer belongs to discrete variable regime and exploits the path information for achieving high dimensions. Thus, we did not cite this work in the previous version of our manuscript. **Thanks to the reviewer, we agree that we should cite this work since the work PhysRevLett.123.070505 is the new advance of quantum teleportation by increasing the dimensions. In the newly resubmitted manuscript, we have cited this reference as a remarkable progress of quantum teleportation.**

For the corresponding revisions, please see point (1) in the list of changes.

Reviewer's Comment:

The "all-optical teleportation" scheme used in this manuscript is based on a theoretical proposal from 1999 which was until now not known to me. It is notably different from conventional teleportation. Rather than sending information on the measurement outcome on Alice's state together with one of the EPR twin photons, here the amplified state itself is sent via a classical channel, seeming closer to "portation" rather than "teleportation". The fidelity that can be reached in this scheme is inherently limited by vacuum noise in addition to detector noise.

While I find the experiment presented in the manuscript appealing, and potentially suitable for publication in Nature Communications, the manuscript is missing

Our Reply:

We thank the reviewer for the deep understanding of the concept of all optical teleportation and mentioning the notable difference of all optical teleportation from

conventional teleportation. We totally agree with the reviewer that all-optical teleportation is closer to "portation" rather than "teleportation" due to its all-optical essence. This is actually a very good point. We are also very glad to see that the reviewer mentions "While I find the experiment presented in the manuscript appealing, and potentially suitable for publication in Nature Communications". We thank the reviewer for the constructive comments and we have thoroughly revised our manuscript according to the reviewer's comments.

For the corresponding revisions, please see points (2-9) in the list of changes.

Reviewer's Comment:

-a clear description of the teleportation scheme.

Our Reply:

We thank the reviewer's thoughtful comment and suggestion for our work. We agree with the reviewer that we should give a clear description of the teleportation scheme to increase the readability of our manuscript. Following the reviewer's comment, we have thoroughly rewritten the section of "OAM Multiplexed AOT Architecture" to give a clear description of the all-optical teleportation scheme in our newly resubmitted manuscript. Such revision has largely improved the readability of our manuscript.

For the corresponding revisions, please see points (2) and (3) in the list of changes.

Reviewer's Comment:

-details on the atomic processes for the EPR pair generation and in particular the operation of the parametric amplifier.

Our Reply:

We thank the reviewer for pointing this out. We agree with the reviewer. We have added the details on the atomic processes for the EPR pair generation and the operation of the parametric amplifier in our newly resubmitted manuscript according to the suggestion of the reviewer.

For the corresponding revisions, please see points (4-7) in the list of changes.

Reviewer's Comment:

-a more thorough theoretical analysis

Our Reply:

We thank the reviewer's carefully reading our paper and pointing this out. We agree with the reviewer that we should give a more thorough theoretical analysis. Such revision will make the manuscript more readable. We have added a thorough theoretical analysis in the section of "OAM Multiplexed AOT Architecture" of our newly resubmitted manuscript following the reviewer's comment.

For the corresponding revisions, please see points (2) and (3) in the list of changes.

Reviewer's Comment:

-an explanation why a continuous variable scheme is required for states described in a subset of a discrete Hilbert space.

Our Reply:

We thank the reviewer for pointing this out. We are sorry for bringing such confusion. Please allow us to clarify this as follows in details.

First of all, the entanglement source for our all-optical teleportation scheme belongs to continuous variable regime. Essentially, our entanglement source consists of a series of parallel two-mode squeezed vacuum states carrying different OAM modes, i.e., OAM multiplexed continuous variable entanglement, which are characterized by amplitude quadrature and phase quadrature of the optical fields [1]. **It is the continuous variable essence of the entanglement source that determines the continuous variable essence of the teleportation in our scheme.**

Secondly, our teleported states are coherent states carrying different OAM modes. **Although the OAM modes themselves are a subset of a discrete Hilbert space, what we teleported is the amplitude quadrature and phase quadrature of the coherent state, which are described by a continuous Hilbert space. From this point of view, our experiment demonstrates a series of parallel continuous variable quantum teleportation of coherent states carrying different OAM modes. In other words, in our scheme, OAM plays a role of realizing multiplexing, similar to taking the advantage of OAM multiplexing in either free space [2] or fiber [3] channels for classical optical communication. Therefore, our system is OAM multiplexed continuous variable quantum teleportation.**

[1] X. Pan, S. Yu, Y. Zhou, K. Zhang, K. Zhang, S. Lv, S. Li, W. Wang, and J. Jing, Phys. Rev. Lett. 123, 070506 (2019).

[2] J. Wang, J.-Y. Yang, I. M. Fazal, N. Ahmed, Y. Yan, H. Huang, Y. Ren, Y. Yue, S. Dolinar, M. Tur, and A. E. Willner, Nat. Photonics 6, 488 (2012).

[3] N. Bozinovic, Y. Yue, Y. Ren, M. Tur, P. Kristensen, H. Huang, A. E. Willner, and S. Ramachandran, Science 340, 1545 (2013).

For the corresponding revisions, please see point (8) in the list of changes.

Reviewer's Comment:

-experimental details on the generation of the state to be teleported.

Our Reply:

We thank the reviewer's carefully reading our paper and pointing this out. We believe that this will also improve the readability of our manuscript. **Following the reviewer's comment, we have added the experimental details on the generation of the state to be teleported in Fig. 2, its caption and the corresponding description in our newly resubmitted manuscript.**

For the corresponding revisions, please see points (6, 7, 9) in the list of changes.

Replies to Reviewer 2

Report of Reviewer 2 -- NCOMMS-20-10407-T

Reviewer's Comment:

The manuscript by S. Liu reports the experimental demonstration of the continuous variable teleportation protocol realised using entangled orbital angular momentum (OAM) modes of light beams. The teleportation scheme implemented is the all-optical quantum teleportation proposed by T. Ralph in 1999. Instead of Bell state measurements and feed-forward, all-optical quantum teleportation requires linear amplification of the input state. In the present experiment, the linear amplifier is replaced by an OAM mode-matched parametric amplifier. The paper shows that teleportation can be established using OAM modes up to the topological charge of 4, beating the classical fidelity limit. The teleportation of OAM mode superpositions has also been demonstrated, showing the possibility to achieve simultaneous teleportation of OAM modes.

The paper is very interesting and I support its publication in Nature Communications; however, I have some comments and questions that should be addressed before final acceptance.

Our Reply:

We thank the reviewer for the patience on carefully reading our manuscript and giving concise and accurate summary for our manuscript. We also thank the reviewer for the valuable and constructive comments on our manuscript. We are very glad to see that the reviewer mentions "The paper is very interesting and I support its publication in Nature Communications". **We have thoroughly revised our manuscript according to the reviewer's comments.**

For the corresponding revisions, please see points (3-4, 6-7 and 10-20) in the list of changes.

Reviewer's comment:

1) In the description of the experiment (Results section), the EPR source should be pumped with a Gaussian beam but is not specified. About this I have a curiosity that could be interesting for clarifying the generation mechanism. Which is the limiting factor in using a smaller waist? Is it useful to improve the parametric gain and the squeezing?

Our Reply:

We thank the reviewer for pointing this out. We are sorry for not specifying that the pump beam is a Gaussian beam. In our scheme, pump beam is a Gaussian beam from a cavity stabilized Ti:sapphire laser. **We have specified this in the newly resubmitted manuscript.**

It is also a very good point to manipulate the system by controlling the pump beam waist. In fact, we have studied this effect in our previous work [1]. We found that

reducing pump waist will reduce the number of OAM multiplexed entanglement simultaneously excited by the four-wave mixing (FWM) process. Then, in order to achieve a large number of OAM multiplexing and build multiple teleportation channels, a large pump waist is required. Therefore, we set the pump waist of FWM process for producing OAM multiplexed EPR entangled state at about 930 μm , which ensures that the number of OAM multiplexed entanglement in our scheme is 11 as shown in Fig. 4 in our newly resubmitted manuscript. In a word, a smaller waist will limit the number of OAM multiplexed entanglement and further limit the number of OAM multiplexed quantum teleportation channels. Such result is consistent with the study of relation between Schmidt number and pump waist in a nonlinear crystal [2].

We would like also to clarify the relation between pump waist and the parametric gain or the squeezing as follows. In principle, smaller pump waist can increase the parametric gain and further increase the squeezing to some extent due to the increasing of the power density and the interaction strength. However, in our current experiment, we can only take large pump waist in order to achieve large number of OAM multiplexing as mentioned above. Therefore, reducing the pump waist might be useful to increase the parametric gain and the squeezing at the expense of the number of OAM multiplexing. This could be our future work for optimizing the quantum teleportation fidelity without consideration of OAM multiplexing.

[1] X. Pan, S. Yu, Y. Zhou, K. Zhang, K. Zhang, S. Lv, S. Li, W. Wang, and J. Jing, Phys. Rev. Lett. 123, 070506 (2019).

[2] C. K. Law, and J. H. Eberly, Phys. Rev. Lett. 92, 127903 (2004).

For the corresponding revisions, please see points (4) and (10) in the list of changes.

Reviewer's Comment:

2) Somehow related to the question above, the EPR source seems to be the main limitation in the maximum topological charge that can be reached for quantum teleportation as shown in Fig 3. The author should explain how the pump beam can influence the EPR correlations between OAM modes.

Our Reply:

We thank the reviewer for this comment. Please allow us to explain this as follows from two aspects including pump power and pump waist.

Firstly, in principle, increasing the pump power can increase the squeezing of EPR correlations between OAM modes. However, in experiment, the squeezing will be maximized when the pump power reaches some certain value, which is limited by the excess noises introduced by both the optical losses and the imperfect measurement mode-matching between the optical modes. We always maximize the squeezing of the system with strong enough pump power. In our experiment, we found that a pump power of 100 mW is strong enough to maximize the squeezing level.

Secondly, the pump waist can influence the maximum topological charge of the EPR

correlations. In general, as mentioned above, increasing pump waist can increase the maximum topological charge of the EPR correlations and further increase the number of OAM multiplexing. This is the reason why we choose a large pump waist of 930 μm which ensures a large number of OAM multiplexing.

In a word, it is this pump beam with strong enough power of 100 mW and large waist of 930 μm , which ensures a sufficient number of OAM multiplexed quantum teleportation channels.

For the corresponding revisions, please see point (4) in the list of changes.

Reviewer's Comment:

3) The local oscillator beam is not described in Fig. 1, some descriptions should also be given along with the text. How it is obtained? How much is the local oscillator power? In which way the its phase is locked?

Our Reply:

We thank the reviewer for this comment. We agree with reviewer that some descriptions about the local oscillator beam should also be given along with the text. The local oscillator beam with a power of 650 μW is obtained by setting up a similar FWM process in the OAM-mode matched parametric amplifier at Alice station, which is a few mm above the current beams. Its pump is split from the pump beam of the OAM-mode matched parametric amplifier, while its seed is split from the beam right after the SLM and before \hat{a}_{in} . In this way, the frequency of local oscillator naturally matches the one of \hat{a}_{out} . In order to measure the amplitude quadrature and phase quadrature of \hat{a}_{out} , we lock the relative phase between local oscillator beam and \hat{a}_{out} to 0 and $\pi/2$ by micro-control unit and proportional-integral-differential circuit, respectively. For clarity, these details are not shown in the Figure. We have added the above descriptions about local oscillator beam in our newly resubmitted manuscript according to the reviewer's comment.

For the corresponding revisions, please see points (11-13) in the list of changes.

Reviewer's Comment:

4) The quantum efficiency of the homodyne detector is not reported.

Our Reply:

We thank the reviewer's carefully reading our paper and pointing this out. Our homodyne detector has a quantum efficiency of 97%. We have specified this number in the newly resubmitted manuscript according to the reviewer's comment.

For the corresponding revisions, please see point (14) in the list of changes.

Reviewer's Comment:

5) The implementation of all-optical quantum teleportation requires a linear amplifier. Instead, here a seeded noncollinear parametric amplifier based on FWM is used. Some

clarification should be given along the main text. It would be useful to insert a comment on the equivalence of both approaches. In which way the fidelity is influenced by the noise added during the amplification process? Is the amplifier pump noise also a limiting factor?

Our Reply:

We thank the reviewer's comments. Please allow us to clarify them as follows.

First of all, as mentioned by C. M. Caves in Part 3, Section B of Physical Review D 26, 1817 (1982) [1], an amplifier satisfying Eq.(3.27a) and Eq.(3.27b), is a linear amplifier. Consistently, the linear amplifier in T. C. Ralph's all-optical teleportation theoretical paper [2] is also described by the same equations. In fact, our FWM amplifier can also be described by the same equations as indicated by other group's works [3]. Moreover, this FWM amplifier has been proved to be a low-noise amplifier near the quantum limit [4]. Therefore, our FWM amplifier can be used to implement the linear amplifier required by all-optical teleportation. **In a word, although our FWM amplifier is a noncollinear parametric amplifier, it is still a low-noise linear amplifier near the quantum limit.**

Secondly, it is a good question how the fidelity is influenced by the noise added during the amplification process. The amplification process will certainly add noise to the input state, and this added noise is transmitted to Bob along with the amplified input state. However, as indicated by Eq. (4) of the newly resubmitted manuscript, this added noise originates from the EPR_2 (one half of the EPR entanglement), rather than the vacuum noise introduced by the usual amplification process. Then, Bob uses a beam splitter to couple the amplified input state with EPR_1 (the other half of the EPR entanglement) as indicated by Eq. (5) of the newly resubmitted manuscript. **On one hand, the noise of the amplified input state is greatly reduced by the appropriate attenuation ratio of the beam splitter. On the other hand, EPR_1 is coupled to the attenuated state through this beam splitter. In this way, the noise cancellation is realized by using the quantum correlation between the two entangled EPR beams, ensuring the beating of the classical limit and thus the realization of the quantum teleportation.** Moreover, as shown in Eq. (8) of the newly resubmitted manuscript, when $G_{2,l} \gg 1$ as required by all-optical teleportation protocol [2], the fidelity of all-optical teleportation protocol is exactly the same as that of Braunstein-Kimble (BK) protocol [5]. Under this condition, the fidelity will be only affected by $G_{1,l}$ i.e., EPR correlation as shown in Eq. (8). In a word, the noise introduced by this amplification process is indispensable for realizing quantum teleportation. **It plays the role of introducing EPR_2 into the system for implementing the noise cancellation operation at Bob station, then makes the fidelity beating the classical limit and thus realizing the quantum teleportation.** In addition to the added noise originating from the EPR_2 as discussed above, there will be also excess noise added due to the experimental imperfection of slight mode mismatching between the input state and EPR_2 . This will introduce uncorrelated vacuum noise into the system and degrade the fidelity slightly.

Thirdly, since the pump beam is very strong in power, it can be regarded as classical

field and remains unaffected by its coupling to the signal and idler. Therefore, the pump noise will not add extra noise into the system and thus will not affect the fidelity of all-optical teleportation.

[1] C. M. Caves, Phys. Rev. D 26, 1817 (1982).

[2] T. C. Ralph, Opt. Lett. 24, 348 (1999).

[3] R. C. Pooser and B. Lawrie, Optica 2, 393 (2015).

[4] R. C. Pooser, A. M. Marino, V. Boyer, K. M. Jones, and P. D. Lett, Phys. Rev. Lett. 103, 010501 (2009).

[5] S. L. Braunstein and H. J. Kimble, Phys. Rev. Lett. 80, 869 (1998).

For the corresponding revisions, please see points (3) and (15-17) in the list of changes.

Reviewer's Comment:

6) The caption of Fig. 1: the notation $|\hat{a}_{in}\rangle$ is not clear, it should be replaced by mode \hat{a}_{in} populated by a coherent state $|\alpha_{in}\rangle$. Fig. 1 should be changed accordingly. There is a double Rb in the caption.

Our Reply:

We thank the reviewer's carefully reading our paper and pointing these notation issues out. We totally agree with the reviewer. We have revised Fig. 1 and the caption of Fig. 1 (Fig. 2 in our newly resubmitted manuscript) according to the suggestion of the reviewer.

For the corresponding revisions, please see points (6-7) in the list of changes.

Reviewer's Comment:

7) It is not clear how the inseparability is calculated. Some clarifications should be given together with references.

Our Reply:

We thank the reviewer for this comment. It will definitely improve the readability of our manuscript. We totally agree with the reviewer that we should insert some clarifications about the calculation of inseparability. We have added this content together with references in our newly resubmitted manuscript.

For the corresponding revisions, please see points (18-19) in the list of changes.

Reviewer's Comment:

8) In Fig.2 the insets of Fig.2c and Fig.2d are not clearly described. The meaning of left, center, and right beams should be explained.

Our Reply:

We thank the reviewer for this comment. We are sorry for not explaining the meaning of these beams. We have explained the meaning of the insets of Fig. 2c and Fig. 2d in the corresponding captions (Fig. 3c and Fig. 3d in our newly resubmitted manuscript).

The left, center, and right beams are $\hat{b}_{1,1}$, Gaussian pump beam, and $\hat{b}_{2,-1}$, respectively.

For the corresponding revisions, please see point (20) in the list of changes.

Replies to Reviewer 3

Report of Reviewer 3 -- NCOMMS-20-10407-T

Reviewer's Comment:

The paper presents the experimental results of an all-optical (OA) teleportation protocol, where a coherent state encoded on an orbital angular momentum (OAM) mode of light ($\ell=1$) is teleported with Fidelity ~ 0.61 (beating the classical limit, i.e., $F=0.5$). Even though the achieved Fidelity is less than the corresponding one achieved through the (conventional) Braunstein-Kimble (BK) protocol, i.e., $F\sim 0.83$ [M. Yukawa et al., Phys. Rev. A 77, 022314 (2008)], this is an important result, since (to my knowledge) that's the first time that the OA teleportation protocol has been experimentally demonstrated, so further progress and higher Fidelities should be expected in the future. The results of this work can potentially make this alternative way to perform teleportation widely known to the community, and other research groups might follow up in the future in order to achieve even higher values of fidelity. Another important result of this work is the experimental demonstration of parallel AO teleportation of two coherent modes (corresponding to different topological charges of OAM), which can be used to to increase the quantum capacity of quantum channels.

The manuscript is well-written and it is worth to be published in Nature Communications for the reasons mentioned before, however, there is a specific claim in the paper regarding the security of the teleportation that I would strongly advise to be removed.

Our Reply:

We thank the reviewer's patience on carefully reading our manuscript and giving concise and accurate summary for our manuscript. We thank the reviewer for high recognition of our work and mentioning that our work is the first time of experimental demonstration of all-optical teleportation. We also thank the reviewer for the valuable and constructive comments on our manuscript. We are glad to see that the reviewer mentions "The manuscript is well-written and it is worth to be published in Nature Communications for the reasons mentioned before,". **We have thoroughly revised our manuscript according to the reviewer's comments. In particular, we have removed the claim regarding the security of the teleportation following the reviewer's comment.**

For the corresponding revisions, please see points (6, 21-28) in the list of changes.

Reviewer's Comment:

In particular, the authors argue that if we block the arm of the entangled state that goes to Bob, then Bob acts as an eavesdropper (Eve), and under this modification, the new measured Fidelity ~ 0.37 lies below the classical limit ($F=0.5$), and thus they show security against eavesdropping. However, that's not the right way of assessing security in a teleportation protocol. Following the discussion of Ref.[F. Grosshans and P. Grangier, Phys. Rev. A 64, 010301(R) (2001)] (or the more recent analysis that connects

the security of the teleportation with the entanglement/steering shared between Alice and Bob in Ref.[Q. He et al., Phys. Rev. Lett. 115, 180502 (2015)], the security of the teleported state is related to Alice's ability to cheat by cloning the initial state. In particular, let us assume that Alice can perfectly perform a quantum cloning protocol, and thus achieve two clones of the initial state with Fidelities of $2/3$ each. Then in principle she can perfectly teleport one of her clones to Bob. However, since Bob doesn't know that Alice cheated, he measures $F=2/3$. Thus, in order Bob to be sure that his state is better than any other potential clones he needs to achieve a fidelity higher than $2/3$ (also known as the no-cloning limit). Another way of looking at the same problem is by assuming that Eve has full access to the losses along at least one arm of the entangled state (along with her access to the classical channel between Alice and Bob). Then Eve's best strategy is to build her own version of the teleported state. If the transmissivity of the channel is below $1/2$ then Eve can create a better copy than Bob, which again means that $F>2/3$ is required for a secure teleportation.

Our Reply:

We thank the reviewer for the very detailed explanation about the security of teleportation. Such explanation is very professional. We totally agree with the reviewer. We have thoroughly removed the claim of the teleportation security in our newly resubmitted manuscript according to the reviewer's comment.

For the corresponding revisions, please see points (21-22) in the list of changes.

Reviewer's comment:

I also have a question regarding the results of the parallel OA teleportation. More specifically, in Fig.4(a) a Fidelity ~ 0.63 is reported for the superposition of modes $\ell=-1$ and $\ell=1$. I find it really weird that this fidelity is higher than the fidelity reported for the single teleported mode $\ell=1$, i.e., $F\sim 0.61$. I would assume that teleporting a superposition of modes is a harder task (as it is actually shown later in Fig.(b)-(d) with Fidelities ranging from ~ 0.54 to ~ 0.6). It would be nice if the authors could provide an explanation for this counter-intuitive result.

Our Reply:

We thank the reviewer for this comment. We are sorry for bringing such confusion and please allow us to clarify this as follows.

First of all, we think this phenomenon is caused by measurement error. It is difficult to keep the experimental conditions exactly the same for every measurement. For example, the tiny fluctuation of the temperature of the vapor cell can cause the slight variation of the final results.

Secondly, in our system, teleporting a superposition of modes is not a harder task compared with teleporting a single mode. From teleporting a single mode to teleporting a superposition of modes, the only thing that needs to be changed is the computer-generated hologram loaded onto the SLM. That is to say, there is no need to realign the optical setup. Because the OAM modes with topological charges $+l$ and $-l$ are

symmetrical in space, the experimental conditions for teleporting a superposition of modes ($+l$ and $-l$) are almost the same as the ones for teleporting a single mode ($+l$ or $-l$). For example, the squeezing levels of the EPR entanglement for a superposition mode and a single mode are almost the same. This has been experimentally shown in our previous work [1]. Therefore, teleporting a superposition of modes is as hard as teleporting a single mode in experiment. The slightly different fidelity results for these two cases from our experiment are caused by measurement error. Again, we thank the reviewer for this comment.

[1] X. Pan, S. Yu, Y. Zhou, K. Zhang, K. Zhang, S. Lv, S. Li, W. Wang, and J. Jing, Phys. Rev. Lett. 123, 070506 (2019).

For the corresponding revisions, please see point (23) in the list of changes.

Reviewer's Comment:

Below, I also have a list of further (minor) suggestions:

1. The first sentence in the abstract "Quantum teleportation is the most essential and fascinating protocol in quantum information." should be toned down a bit, and rewritten as "one of the most essential..."

Our Reply:

We thank the reviewer for pointing this out. We totally agree with the reviewer. We have revised this sentence in our newly resubmitted manuscript according to the reviewer's suggestion.

For the corresponding revisions, please see point (24) in the list of changes.

Reviewer's Comment:

2. Right after Eq.(2) the authors should clearly state that the two last terms in Eq.(2) vanish in the limit of an EPR entangled state and very high parametric gain, leaving as an output state only the first term, which is equal to the input state, thus we achieve teleportation.

Our Reply:

We thank the reviewer for the valuable and thoughtful suggestion for our work. We totally agree with reviewer that we should clearly state the above description suggested by the reviewer. This will greatly improve the readability of our manuscript. We have added the above description in our newly resubmitted manuscript following the reviewer's suggestion.

For the corresponding revisions, please see point (25) in the list of changes.

Reviewer's Comment:

3. In Fig.3 a comparison between the established entanglement and the achieved fidelity for the different OAM modes is presented. It is true that in the literature the separability criterion (or the logarithmic negativity, which is a monotonic function of the maximum

possible violation of the separability criterion) has been extensively used to "measure" entanglement, but there are more rigorous ways to quantify entanglement, such as: (i) entanglement of formation, (ii) relative entropy of entanglement, and (iii) squashed entanglement.

Our Reply:

We thank the reviewer for mentioning the other more rigorous ways to quantify entanglement. Please allow us to explain the reason for choosing separability criterion.

In our scheme, the fidelity of all-optical teleportation is directly related to the two-mode squeezing of the OAM multiplexed entanglement. The separability criterion is also directly related to the two-mode squeezing of OAM multiplexed entanglement. This is the reason why we use separability criterion for characterizing the entanglement here. We totally agree with the reviewer that entanglement of formation, relative entropy of entanglement, and squashed entanglement are more rigorous ways to quantify entanglement. This will be definitely our future studies.

For the corresponding revisions, please see point (26) in the list of changes.

Reviewer's Comment:

4. I think that the authors should explain the reasoning behind picking a half wave plate to represent the all-optical channel between Alice and Bob.

Our Reply:

We thank the reviewer for carefully reading our paper and pointing this out. We are very sorry for bringing such confusion. This half wave plate is just used to change the polarization of the beam from horizontal to vertical. **It is better to put it right before the first PBS in Bob station and therefore we have moved this half wave plate to Bob station in Fig.2 of our newly resubmitted manuscript.**

For the corresponding revisions, please see point (6) in the list of changes.

Reviewer's Comment:

5. The papers [M. Yukawa et al., Phys. Rev. A 77, 022314 (2008)] and [N. Lee et al., Science 332, 330 (2011)] should be cited in the introduction regarding previous works on experimental teleportation results, since they present the current state of the art on BK teleportation experiments.

Our Reply:

We thank the reviewer for this suggestion. We totally agree with the reviewer that these works [M. Yukawa et al., Phys. Rev. A 77, 022314 (2008)] and [N. Lee et al., Science 332, 330 (2011)] present the current state of the art on BK teleportation experiments. **We have cited these two papers in our newly resubmitted manuscript following the reviewer's suggestion.**

For the corresponding revisions, please see points (27, 28) in the list of changes.

List of Changes:

- (1) We have added
“[19] Luo, Y.-H. et al. Quantum teleportation in high dimensions. Phys. Rev. Lett. 123, 070505 (2019).” in the reference part.
- (2) We have thoroughly rewritten the section of “OAM Multiplexed AOT Architecture”.
- (3) We have added
“[45] Pooser, R. C., Marino, A. M., Boyer, V., Jones, K. M., & Lett, P. D. Low-Noise Amplification of a Continuous-Variable Quantum State. Phys. Rev. Lett. 103, 010501 (2009)
[47] Caves, C. M. Quantum limits on noise in linear amplifiers. Phys. Rev. D 26, 1817-1839 (1982).
[48] Braunstein, S. L., Fuchs, C. A., Kimble, H. J. & van Loock, P. Quantum versus classical domains for teleportation with continuous variables. Phys. Rev. A 64, 022321 (2001).” in the reference part.
- (4) We have added the sentences
“One beam, which is vertically polarized and has a power of 100 mW, is served as the pump beam of the FWM process in a ^{85}Rb vapor cell for producing OAM multiplexed EPR entangled state³⁹. Such pump power is strong enough to maximize the squeezing of our system. As shown in Fig. 2b, in this double-A configuration FWM process, two pump photons convert to one photon for EPR_1 (blue-shifted from the pump beam) and one photon for EPR_2 (red-shifted from the pump beam). Reflected by a Glan-Laser polarizer (GL), this pump beam is seeded into the ^{85}Rb vapor cell which is 12 mm long and stabilized at 113 °C. This pump beam has a large waist of about 930 μm at the center of vapor cell. This pump beam with strong enough power of 100 mW and large waist of 930 μm ensures a sufficient number of OAM multiplexed quantum AOT channels³⁹. The residual pump beam after the FWM process is eliminated by a Glan-Thompson polarizer (GT) with an extinction ratio of $10^5:1$. In this way, we obtain the OAM-multiplexed EPR entangled state for realizing quantum AOT.” in the fifth paragraph.
- (5) We have added the sentences
“This OAM mode-matched PA is also based on the double-A configuration FWM process in another ^{85}Rb vapor cell which is 12 mm long and stabilized at 110 °C. Alice amplifies the input state \hat{a}_{in} carrying OAM modes through this OAM mode-matched PA with the help of EPR_2 . Combined by a GL, the pump, input state \hat{a}_{in} , and EPR_2 are crossed in the center of the ^{85}Rb vapor cell. Due to OAM conservation, only the input state \hat{a}_{in} and EPR_2 with opposite topological charges can be coupled when the pump is a Gaussian beam. The angle between input state \hat{a}_{in} and EPR_2 is about 14 mrad and the pump beam is symmetrically crossed with \hat{a}_{in} and EPR_2 beams in the same plane.” in the fifth paragraph.

(6) We have redrawn the Fig. 2 (Fig. 1 in the first submission).

(7) We have revised the caption of Fig. 2 (Fig. 1 in the first submission) as

“Detailed experimental setup for parallel AOT by multiplexing OAM channels. a, Alice, Bob, and Victor act as sender, receiver, and verifier of the AOT, respectively. Alice and Bob are connected by an all-optical channel. $\hat{\alpha}_{in}$, input state populated by a coherent state $|\alpha_{in}\rangle$; EPR state, Einstein-Podolsky-Rosen entangled state; ^{85}Rb , vapor cell; HWP, half wave plate; PZT, piezoelectric transducer; PBS, polarization beam splitter; AOM, acousto-optic modulator; SLM, spatial light modulator; GL, Glan-Laser polarizer; GT, Glan-Thompson polarizer; $\hat{\alpha}_{out}$, retrieved state; OS, oscilloscope; SA, spectrum analyzer. The resolution bandwidth (RBW) of the SA is 1 MHz. The video bandwidth (VBW) of the SA is 100 Hz. b, Energy level diagram of ^{85}Rb D1 line for FWM process in our scheme. Δ , one-photon detuning; δ , two-photon detuning.”

(8) We have replaced the sentence

“These results clearly show that we have experimentally and deterministically constructed 9 OAM multiplexed quantum AOT channels with fidelities beating the classical limit.”

with

“Although the OAM modes themselves are a subset of a discrete Hilbert space, what we teleported is the amplitude quadrature and phase quadrature of the coherent state, which are described by a continuous Hilbert space. From this point of view, our experiment demonstrates 9 parallel channels of deterministic CV quantum teleportation of coherent states carrying different OAM modes.” in the first sentence of the last paragraph.

(9) We have added the sentences

“The other beam from the first PBS, which is horizontally polarized, is also divided into two by another PBS. The weak one passes through an acousto-optic modulator (AOM) and a spatial light modulator (SLM) to generate the OAM-mode coded input state $\hat{\alpha}_{in}$ which is blue-shifted about 3.04 GHz from the pump beam and has a power of 0.4 μW .” in the fifth paragraph.

(10) We have added the sentence

“A cavity stabilized Ti:sapphire laser emits a Gaussian laser beam.” in the fifth paragraph.

(11) We have added the sentences

“The LO beam with a power of 650 μW is obtained by setting up a similar FWM process in the OAM-mode matched PA at Alice station, which is a few mm above

*the current beams*⁴³. Its pump is split from the pump beam of OAM-mode matched PA, while its seed is split from the beam right after the SLM and before \hat{a}_{in} . In this way, the frequency of LO beam naturally matches the one of \hat{a}_{out} .” in the fifth paragraph.

(12) We have added

“[50] Huang, K., Le Jeannic, H., Ruaudel, J., Morin, O., & Laurat, J. Microcontroller-based locking in optics experiments, Rev. Sci. Instrum. 85, 123112 (2014).” in the reference part.

(13) We have replaced the sentence

“The amplitude and phase quadrature variances of the teleported state carrying OAM mode with $l=1$ measured by Victor’ balanced homodyne detection (BHD) at 2 MHz sideband are shown in Fig. 2a (locking BHD phase to 0) and Fig. 2b (locking BHD phase to $\pi/2$), respectively.”

with

“The amplitude and phase quadrature variances of the teleported state carrying OAM mode with $l=1$ measured by Victor’s BHD at 2 MHz sideband are shown in Fig. 3a (locking the relative phase between LO beam and \hat{a}_{out} to 0 by micro-control unit^{49,50}) and Fig. 3b (locking the relative phase between LO beam and \hat{a}_{out} to $\pi/2$ by proportional-integral-differential circuit⁴⁹), respectively.” in the beginning of “Teleporting a single OAM mode” section.

(14) We have added the sentence

“Our balanced homodyne detector has a transimpedance gain of 10^5 V/A and a quantum efficiency of 97%.” in the fifth paragraph.

(15) We have added the sentence

“According to the definition given in other works^{4,47}, this OAM mode-matched PA based on double- Λ configuration FWM process⁴⁵, as described by Eq.(4), is a linear amplifier.” in the third paragraph.

(16) We have added the sentences

“In other words, although the amplification process will certainly add noise to the input state as indicated by Eq.(4), such added noise will be greatly reduced by the appropriate attenuation ratio of the beam splitter at Bob station and the coupling of two entangled EPR beams on the same beam splitter as shown by Eq.(5). In this way, the noise cancellation is realized by using the quantum correlation between the two entangled EPR beams, thus ensuring the realization of quantum teleportation. In a word, the noise introduced by this amplification process is indispensable for realizing quantum AOT. It plays the role of introducing EPR_2 into the system for implementing the subsequent noise cancellation operation at Bob station, then makes the fidelity beating the classical limit and thus realizing the quantum AOT.” in the third paragraph.

(17) We have added the sentences

“Since the pump beam is very strong in this FWM process, it can be regarded as classical field. Therefore, the pump noise will not add extra noise into the system. In this way,” in the third paragraph.

(18) We have added

“[52] Duan, L. M., Giedke, G., Cirac, J. I., & Zoller, P. Inseparability Criterion for Continuous Variable Systems. Phys. Rev. Lett. 84, 2722-2725 (2000).”

[53] Simon, R. *Peres-Horodecki separability criterion for continuous variable systems.* *Phys. Rev. Lett.* 84, 2726-2729 (2000).

[54] Takei, N., Yonezawa, H., Aoki, T. & Furusawa, A. *High-fidelity teleportation beyond the no-cloning limit and entanglement swapping for continuous variables.* *Phys. Rev. Lett.* 94, 220502 (2005).” in the reference part.

(19) We have added the sentence

“*We calculate the inseparability of OAM-multiplexed entanglement source by $I_{\hat{b}_{1,l},\hat{b}_{2,-l}} = \text{Var}(\hat{X}_{\hat{b}_{1,l}} - \hat{X}_{\hat{b}_{2,-l}}) + \text{Var}(\hat{Y}_{\hat{b}_{1,l}} + \hat{Y}_{\hat{b}_{2,-l}})$* ⁵²⁻⁵⁴” in the seventh paragraph.

(20) We have added the sentence

“*The left, center, and right beams are $\hat{b}_{1,1}$, Gaussian pump beam, and $\hat{b}_{2,-1}$, respectively.*” in the caption of Fig. 3 (Fig. 2 in the first submission).

(21) We have replaced the sentences

“*If we block the EPR_1 , Bob acts as an eavesdropper (Eve). The quadrature variances of the state retrieved by Eve (yellow traces in Fig. 2a and Fig. 2b) are 6.44 ± 0.14 dB above the corresponding quadrature variances of the input state. In such case, the fidelity of the retrieved state is only 0.37 ± 0.01 . This is significantly lower than both the classical limit and the fidelity of the quantum AOT with the help of EPR entanglement, showing the security of the AOT against eavesdropping.*”

with

“*If we block EPR_1 , the quadrature variances of the state retrieved by Bob (yellow traces in Fig. 3a and Fig. 3b) are 6.44 ± 0.14 dB above the corresponding quadrature variances of the input state. In such case, the fidelity of the retrieved state is only 0.37 ± 0.01 . This is significantly lower than both the classical limit and the fidelity of the quantum AOT with the help of EPR entanglement, showing the importance of the EPR entanglement for realizing quantum AOT.*” in the sixth paragraph.

(22) We have replaced the sentences

“*The yellow dot trace in Fig. 3a is the fidelity of AOT with the help of EPR_2 and a vacuum state for Eve. It is always lower than both the red and the blue dot traces, further confirming the security of our OAM multiplexed AOT against eavesdropping.*”

with

“*The yellow dot trace in Fig. 4a is the fidelity of AOT with the help of EPR_2 and a vacuum state. It is always lower than both the red and the blue dot traces, further confirming the importance of the EPR entanglement for realizing quantum AOT.*” in the seventh paragraph.

(23) We have added the sentences

“*In principle, the fidelity for teleporting $\hat{a}_{in} = \hat{a}_{in,l} + \hat{a}_{in,-l}$ is same as the one for teleporting $\hat{a}_{in} = \hat{a}_{in,l}$ ($\hat{a}_{in,-l}$) because the squeezing levels of EPR entanglement for these two cases are equal³⁹. In our experiment, the slight difference between the fidelities for these two cases is caused by measurement error.*” in the eighth paragraph.

(24) We have replaced the sentence

“Quantum teleportation is the most essential and fascinating protocol in quantum information.”

with

“Quantum teleportation is one of the most essential and fascinating protocol in quantum information.” in the abstract.

(25) We have added the sentence

“The two last terms in Eq.(5) vanish in the limit of an EPR entangled state ($G_{1,l} \gg 1$) and very high parametric gain ($G_{2,l} \gg 1$), leaving as an output state only the first term, which is equal to the input state $\hat{a}_{in,l}$, thus we achieve teleportation.”

in the third paragraph.

(26) We have added the sentence

“We calculate the inseparability of OAM-multiplexed entanglement source by $I_{\hat{b}_{1,l}, \hat{b}_{2,-l}} = \text{Var}(\hat{X}_{\hat{b}_{1,l}} - \hat{X}_{\hat{b}_{2,-l}}) + \text{Var}(\hat{Y}_{\hat{b}_{1,l}} + \hat{Y}_{\hat{b}_{2,-l}})$ ⁵²⁻⁵⁴, which is directly related to the two-mode squeezing of OAM multiplexed entanglement.” in the seventh paragraph.

(27) We have cited references [20], [21] in the first paragraph.

(28) We have added

“[20] Yukawa, M., Benichi, H. & Furusawa, A. High-fidelity continuous-variable quantum teleportation toward multistep quantum operations. Phys. Rev. A 77, 022314 (2008).

[21] Lee, N. et al. Teleportation of nonclassical wave packets of light. Science 332, 330-333 (2011).” in the reference part.

(29) We have revised “Supplemental material” correspondingly.

Reviewers' Comments:

Reviewer #1:

Remarks to the Author:

The manuscript is much improved and I am happy to support publication now.

Reviewer #2:

Remarks to the Author:

The authors have extensively replied and answered to all my comments and questions. The paper has been modified accordingly. I strongly support the publication of this paper in Nature Communications.

Reviewer #3:

Remarks to the Author:

I would like to thank the authors for their comments. I'm happy with their responses which clarified the questions I had on their work. The revised paper looks good to me to proceed for publication, and I have no further remarks apart from a minor suggestion (see below) that it's up to the authors to decide if they want to implement it:

In lines 273-281 there is a revised discussion on the security analysis. There is nothing wrong with the revised text, but I just don't see if it's any helpful anymore. I mean that since the EPR₁ signal that goes to Bob is blocked but Alice still gets the EPR₂ one, it is a bit obvious that the fidelity will drop significantly since it's always better for Alice to mix her state with a vacuum instead of a thermal state if no entanglement with Bob is involved.

Response Letter

Response to the Reviewers:

Replies to Reviewer 1

Reviewer's Comment:

The manuscript is much improved and I am happy to support publication now.

Our Reply:

We are glad to see that reviewer mentions “The manuscript is much improved and I am happy to support publication now”.

Again, we thank reviewer for all the constructive comments and useful suggestions through the entire review process, which have greatly improved the readability of the manuscript and the quality of our work.

Replies to Reviewer 2

Reviewer's Comment:

The authors have extensively replied and answered to all my comments and questions. The paper has been modified accordingly. I strongly support the publication of this paper in Nature Communications.

Our Reply:

We thank reviewer's high recognition of our reply. We are also glad to see that reviewer mentions "I strongly support the publication of this paper in Nature Communications".

Again, we thank reviewer for all the constructive comments and useful suggestions through the entire review process, which have greatly improved the readability of the manuscript and the quality of our work.

Replies to Reviewer 3

Reviewer's Comment:

I would like to thank the authors for their comments. I'm happy with their responses which clarified the questions I had on their work. The revised paper looks good to me to proceed for publication, and I have no further remarks apart from a minor suggestion (see below) that it's up to the authors to decide if they want to implement it:

In lines 273-281 there is a revised discussion on the security analysis. There is nothing wrong with the revised text, but I just don't see if it's any helpful anymore. I mean that since the EPR₁ signal that goes to Bob is blocked but Alice still gets the EPR₂ one, it is a bit obvious that the fidelity will drop significantly since it's always better for Alice to mix her state with a vacuum instead of a thermal state if no entanglement with Bob is involved.

Our Reply:

We thank reviewer for the valuable and constructive comment on our manuscript. We are glad to see that reviewer mentions “The revised paper looks good to me to proceed for publication”. We have removed the part about all-optical teleportation with the help of EPR₂ and a vacuum state in our newly submitted manuscript according to reviewer's suggestion.

Again, we thank reviewer for all the constructive comments and useful suggestions through the entire review process, which have greatly improved the readability of the manuscript and the quality of our work.

For corresponding revisions, please refer to the following list of changes.

List of Changes:

(1) We have redrawn the Fig. 3 and Fig. 4.

Fig. 3

Fig. 4

(2) We have removed the sentences

“If we block EPR₁, the quadrature variances of the state retrieved by Bob (yellow traces in Fig. 3a and Fig. 3b) are 6.44 ± 0.14 dB above the corresponding quadrature variances of the input state. In such case, the fidelity of the retrieved state is only 0.37 ± 0.01 . This is significantly lower than both the classical limit and the fidelity of the quantum AOT with the help of EPR entanglement, showing the importance of the EPR entanglement for realizing quantum AOT.”

(3) We have removed the sentences

“The yellow dot trace in Fig. 4a is the fidelity of AOT with the help of EPR₂ and a vacuum state. It is always lower than both the red and the blue dot traces, further confirming the importance of the EPR entanglement for realizing quantum AOT.”